# Towards triple elimination of HIV, syphilis and HBV mother-to-child transmission: Protocol of a simplified and integrated strategy in Burkina Faso and The Gambia: Protocol for the phase 1 of the TRI-MOM project

Erwan Vo-Quang[1,2*☯], Alice N. Guingané[3☯], Lauren Périères[4☯], Gibril Ndow[1], Victor Some[5], Sheriff Badjie[6], Asta Jobe[7], Clarisse Gouem[8], Yusuke Shimakawa[9], Dramane Kania[8], Maud Lemoine[1,10☯], Sylvie Boyer[4☯], on behalf of the TRI-MOM study group[¶]

**1** MRC The Gambia Unit@LSHTM, Banjul, The Gambia, **2** Team "Viruses, Hepatology, Cancer", INSERM U955, Créteil, France, **3** Département d'Hépato-Gastroentérologie, Centre Hospitalier Universitaire Yalgado Ouédraogo, Ouagadougou, Burkina Faso, **4** Aix Marseille Univ, INSERM, IRD, SESSTIM, Sciences Economiques & Sociales de la Santé & Traitement de l'Information Médicale, ISSPAM, Marseille, France, **5** REVS PLUS, Bobo-Dioulasso, Burkina Faso, **6** Viral Hepatitis Focal Point at National AIDS Control Program, The Gambia Ministry of Health & Social Welfare, Banjul, Gambia, **7** Young Gambian Mums Fund, Banjul, The Gambia, **8** Centre Muraz, Institut National de Santé Publique, Bobo-Dioulasso, Burkina Faso, **9** Unité d'Épidémiologie des Maladies Émergentes, Institut Pasteur, Paris, France, **10** Department of Metabolism, Digestion and Reproduction, Division of Digestive Diseases, Imperial College London, London, United Kingdom

¶ Membership of the TRI-MOM study group is listed in the Acknowledgments.
☯ These authors contributed equally to this work.
* erwan.voquang@gmail.com

## Abstract

### Introduction

Mother-to-child transmission (MTCT) of HIV, syphilis, and hepatitis B virus (HBV) commonly observed in the WHO African region is associated with excess morbidity and mortality. Despite some progress, the coverage of interventions to prevent MTCT of these infections remains insufficient, particularly for syphilis and HBV. To fulfil these gaps and achieve the triple elimination of MTCT of these infections by 2030, the World Health Organization (WHO) advocates for integration of prevention of MTCT (PMTCT) activities for HBV with HIV and syphilis antenatal services. In partnership with the local governments, the TRI-MOM project, conducted in 2 phases, aims to evaluate a simplified (based on inexpensive rapid diagnostic tests), integrated (in maternal and child health services) and coordinated (between the various programs and health care workers) strategy for the triple elimination of HIV, syphilis and HBV MTCT in Burkina Faso and The Gambia.

**Data availability statement:** No datasets were generated or analysed during the current study. All relevant data from this study will be made available upon study completion.

**Funding:** This study is financially supported by L'Initiative, implemented by Expertise France, in the form of a grant awarded to SB (22-SB2712). This study is also financially supported by the European Association for the Study of the Liver in the form of a Juan Rodés PhD Studentship awarded to EVQ. The funders had no role in study design, data collection and analysis, decision to publish, or preparation of the manuscript.

**Competing interests:** I have read the journal's policy and the authors of this manuscript have the following competing interests: GN, and YS report research funding and consultancy fees from Gilead Sciences. ML reports consultancy fees from Abbott USA and Gilead Sciences. This does not alter our adherence to PLOS ONE policies on sharing data and materials. All other authors declare no competing interests.

## Methods and analysis

The strategy will be implemented in 5 rural and urban health facilities in each country and will include four activities: i) training sessions for healthcare workers working in maternal and child health services, ii) screening of pregnant women of the three infections using rapid diagnostic tests at the first antenatal visit, iii) clinical assessment and treatment of women tested positive for any of the 3 infections, and iv) raising awareness on HIV, Syphilis and HBV PMTCT among pregnant women and empowering those screened positive. 17,000 pregnant women are expected to be screened. The strategy will be evaluated through an interdisciplinary, mixed-methods approach comprising three studies: i) a quantitative and qualitative cross-sectional study conducted both before and after the implementation of the strategy to assess its impact on triple screening coverage in pregnant women; ii) a an intervention study with longitudinal follow-up of pregnant women positive for any of the three infections to assess the coverage of PMTCT measures; and iii) a cost and cost-effectiveness analysis of the project compared to the reference situation in each country, which will rely on a micro costing study to estimate the incremental cost of the strategy per mother/child couple compared with the reference situation in each country, and compare it to the number of avoided infections.

## Ethics and dissemination

The study protocol has been approved by the competent authorities of the countries participating to the research (the LSHTM/MRCUG Scientific Coordinating Committee, the Gambia Government/MRC Joint Ethics Committee, the LSHTM ethics committee, the Burkinabe National Ethical Committee for Research in Health and the French Commission on Information Technology and Liberties). Results on the feasibility and acceptability of the triple elimination strategy will be disseminated using different media including policy briefs, posters and articles.

## Trial registration number

ClinicalTrials.gov [NCT05951751]().

## Introduction

Sub-Saharan Africa (SSA) is the region with the highest prevalence of human immunodeficiency virus (HIV), syphilis and hepatitis B virus (HBV) infections worldwide [1]. In SSA, MTCT plays a central role in the persistence and dynamic of these 3 epidemics despite the existence of effective prophylactic treatments. In 2024, 56% of pregnant women acquiring HIV received ART, leaving almost half of the babies exposed to a 5–15% risk of transmission [2] instead of less than 2% [3], while HIV MTCT is associated with an increased risk of neonatal mortality and long-term adverse health effects among infected children [4]. MTCT is the second most

common mode of HBV transmission after horizontal transmission during the first years of life [5], whose risk is significantly reduced by the administration of a timely hepatitis B birth dose (HepB-BD) within 24 hours of birth. Unfortunately, the coverage for this vaccine is still insufficient in SSA, and a risk of transmission remains for mother with high viral load, justifying the importance of preventing PMTCT [6] via antiviral therapy with daily tenofovir disoproxyl fumarate (TDF) in highly viraemic pregnant women [7]. Lastly, MTCT of syphilis can lead to dramatic complications, including spontaneous abortions, premature births, neonatal mortality and congenital syphilis [8] though penicillin treatment in pregnant women reduces the risk of congenital syphilis and neonatal death by 97% and 80%, respectively [9]. Furthermore, co-infection with these infections is common among pregnant women and increases the risk of MTCT, neonatal complications and infant mortality [10,11], making the case for a common agenda and for an integrated strategy of the WHO for their elimination [7]. In 2022, the WHO approved a new Global Health Sector Strategies calling for the eMTCT of these 3 infections by 2030 [1]. The progression and validation of elimination of MTCT of HIV, syphilis and HBV is monitored through target values for a set of indicators of impact in terms of new infections, screening coverage, maternal treatment and infant HBV vaccination [12].

Aligned with this strategy, both Burkina Faso and The Gambia have approved a plan for the elimination of MTCT of HIV, syphilis and Hepatitis B, but are still far the validation of the various criteria. The prevalence of the three infections in adults is estimated at 1.3% and 1.8% for HIV [13,14], 1.8% and 6.8% for syphilis [15,16], and 8% and 8% for HBV [17,18], respectively. In the Gambia, the rate of MTCT of HBV has been estimated at 2.8%, higher than the target rate of 0.5% [19]. While both countries have implemented routine screening for HIV and syphilis in pregnant women, access to treatment for positive women is inconsistent, especially for Syphilis [20]. The screening of pregnant women for HBV infection is rarely performed in routine. Viral load measures are never free of charge and rarely available in antenatal services, so that many women are referred to another facility with the adequate equipment. As a result, few positive women receive a prophylactic antiviral therapy.. In The Gambia, the HepB-BD vaccine has been integrated into the national immunisation programme since the early 1990s, following a universal strategy. However, the actual coverage remains low (below 20%, Ndow et al. unpublished data) 36% only among infants born to HBV-infected mothers, [19]. Burkina Faso adopted the universal administration of the HepB-BD vaccine in 2022, but no data is available about the coverage at the beginning of the project.

To date, evidence is scarce about regarding the feasibility of the WHO strategy for elimination of MTCT [21,22], and, to the best of our knowledge, null in settings from SSA. In 2021, Thompson et al. assessed the feasibility of integrating HBV screening into HIV PMTCT services in the Democratic Republic of the Congo, but without including screening for syphilis [23]. Another recent study conducted in Mozambique found that the implementation of triple screening among pregnant women attending antenatal services was associated with a risk of HBV MTCT rate estimated at 0.7% [24]. However, this study was conducted in a single hospital in the capital city and did not assess the impact of the strategy on HIV and syphilis MTCT. Moreover, these studies focus on the clinical outcomes, and leave behind the implementation and socio-economic aspects of their intervention, despite their crucial role as facilitators or barriers to their success. From the healthcare workers perspective, these include knowledge about the three infections and acceptability of the new tasks linked to the strategy [25]. From the pregnant women perspective, they include overcoming the potential stigma or other obstacles associated to the infections to accept screening and adhere to treatment if needed [26,27]. From the healthcare perspective, they encompass the incremental cost of the strategy compared to its non-implementation, with regards to the gains in terms of avoided infections.

## Study objectives

The main objective of the TRI-MOM project is to evaluate a simplified and integrated strategy to support the triple elimination of HIV, syphilis and HBV MTCT in West Africa. The intervention will be implemented in Burkina Faso and The Gambia and will use an innovative approach that will be: i) simplified, i.e., based on inexpensive rapid diagnostic tests (RDT), ii)

integrated into the maternal and child health services, where pregnant women are usually followed up, and iii) coordinated between the various programs and stakeholders involved in the PMTCT of the three infections.

The specific objectives of the TRI-MOM project (Table 1) are:

1. To evaluate the impact of the TRI-MOM strategy on triple screening coverage in pregnant women

2. To assess the whole antenatal care cascade following a positive maternal screening for any of the three infections and the HepB-BD vaccine coverage among their newborns

3. To identify facilitators and barriers associated with the uptake to PMTCT measures (screening, treatment and infant vaccination)

4. To document the health capabilities and vulnerabilities of women of childbearing age

5. To assess the acceptability of the strategy by pregnant women and their partners as well as by the key stakeholders involved in PMTCT in both countries

6. To assess the costs and cost-effectiveness of the strategy.

### Hypothesis

We hypothesise that the implementation of a simplified, integrated and coordinated strategy for triple elimination is feasible and will improve:

i) the geographical and financial accessibility of PMTCT services, by providing PMTCT interventions free of charge within local maternal and child health services and optimizing service organization to reduce costs effectively,

ii) the quality and acceptability of services, by offering comprehensive care focused on women's needs, and

iii) women's literacy and empowerment. These various effects are expected to improve screening coverage in all pregnant women and the coverage of PMTCT measures among those screened positive.

Consequently, we anticipate a significant reduction in perinatal infections and an overall improvement in women and children's health. The effects and results of the strategy will be evaluated using the theory of change framework (Fig 1).

## Methods and analysis

- Study design

**Table 1. Objectives and Evaluation Framework.**

| Specific objectives | Component of the evaluation |
|---|---|
| 1. To evaluate the impact of the TRI-MOM strategy on triple screening coverage in pregnant women; | intervention study with longitudinal follow-up + mixed method study |
| 2. To assess the whole antenatal care cascade following a positive maternal screening for any of the three infections and the HepB-BD vaccine coverage among their newborns; | intervention study with longitudinal follow-up |
| 3. To identify facilitators and barriers associated with the uptake to PMTCT measures (screening, treatment and infant vaccination); | mixed method study |
| 4. To document the health capabilities and vulnerabilities of women of childbearing age; | mixed method study |
| 5. To assess the acceptability of the strategy by pregnant women and their partners as well as by the key stakeholders involved in PMTCT in both countries; | mixed method study |
| 6. To assess the costs and cost-effectiveness of the strategy. | micro-costing study |

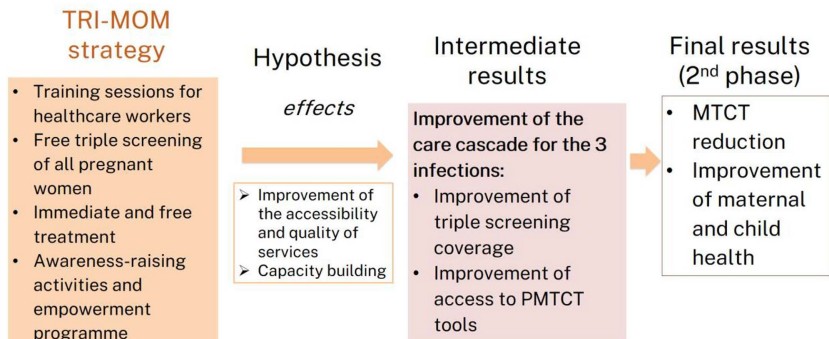

**Fig 1. Theory of change: expected effects and results of the triple elimination strategy (TRI-MOM).** PMTCT: prevention of mother-to-child transmission, MTCT: mother-to-child transmission.

The TRI-MOM project is an intervention study among pregnant and babies born to infected women implemented in 10 sites in Burkina Faso and The Gambia. The evaluation of the intervention will use a multidisciplinary approach including three nested studies: i) a cross-sectional mixed-methods study conducted before and after the implementation of the strategy, ii) a intervention study with longitudinal follow-up of pregnant women tested positive for any of the three infections, and iii) a cost and cost-effectiveness study.

- Study sites

The TRI-MOM strategy will be implemented in 5 maternal and child health services in Burkina Faso and 5 in The Gambia. The study sites have been selected following discussion with both ministries of health and patients' associations to represent the diversity of the local healthcare services: the study sites are located at various levels of the health system (primary care; district level and regional level), in diverse settings (urban and rural) and have varying attendance rates. These health facilities provide antenatal and post-natal services led by either doctors or nurses and have laboratories with different levels of equipment (Table 2).

- Intervention

The triple elimination strategy encompasses four complementary and synergistic activities aimed at promoting universal access to PMTCT for the three targeted infections:

- (i) training sessions for healthcare workers involved in maternal and child health. These sessions will cover screening, pre- and post-test counselling, and PMTCT of the three infections.

- (ii) free triple screening (HIV, syphilis and HBV) of all pregnant women attending their first antenatal consultation. Screening will be performed using RDT, either the Antenatal Care Panel HIV, syphilis, hepatitis B surface antigen (HBsAg) (Abbott, USA) or SD BIOLINE™ HIV/syphilis Duo test and Determine™ HBsAg (S1 Fig). Results will be delivered on site within approximately 15–30 minutes, followed by post-test counselling.

- (iii) in case of positive screening, immediate and free treatment for the infection(s). Treatment includes ART for HIV and HBV co-infection, antibiotic treatment for syphilis, and nucleotide analogue (TDF) for HBV mono-infection (300 mg per day until the 2nd month of postpartum), combined with HepB-BD. Treatment will be delivered immediately (the same day as the screening), on-site according to WHO and national guidelines [7,28,29]. For HBV mono-infection, eligibility for TDF will be determined based on HBV viral load (VL ≥ 200,000 IU/l) measurement using the GeneXpert platform

**Table 2. Main characteristics of study sites in The Gambia and Burkina Faso (TRI-MOM project).**

| Health care facility | Level in the healthcare system | Area | Number of 1st ANC visits and deliveries per month | Access to HBV GeneXpert |
|---|---|---|---|---|
| **Burkina Faso** | | | | |
| Centre Medical (CM) de Gounghin 7 | District | Urban | 100/53 | Yes |
| Centre de santé et de promotion sociale (CSPS) Urbain | Primary | Semi-urban | 40/30 | No |
| CM REVS PLUS | District | Urban | 10/0 | No |
| CSPS Farakan | Primary | Urban | 110/90 | No |
| CSPS Lafiabougou | Primary | Urban | 140/150 | No |
| **The Gambia** | | | | |
| Bundung Maternal & Child Health Hospital | Hospital (tertiary) | Urban | 400/350 | Yes |
| Banjulinding Health Centre | Minor health centre (secondary) | Urban | | |
| Brikama Health Hospital | District hospital (tertiary) | Urban | 400/350 | Yes |
| Bwiam General Hospital | Hospital (tertiary) | Semi-rural | 125/100 | Yes |
| Bureng District Health Centre | Minor health centre (secondary) | Rural | 109/109 | No |

Abbreviation: ANC: antenatal care, HBV: hepatitis B virus CM: Centre Medical. CSPS: Centre de santé et de promotion sociale

where available. If there is no access to HBV viral load, eligibility will be based solely on HBsAg results, as recommended by the revised 2024 WHO guidelines [7]. Treatment initiation algorithms are summarised in Fig 2.

- (iv) awareness-raising activities on PMTCT among pregnant women attending antenatal services coupled with empowerment activities for women who test positive. The empowerment programme aims equip women with the information and support to make free and informed decisions about their health and that of their child. This approach includes a

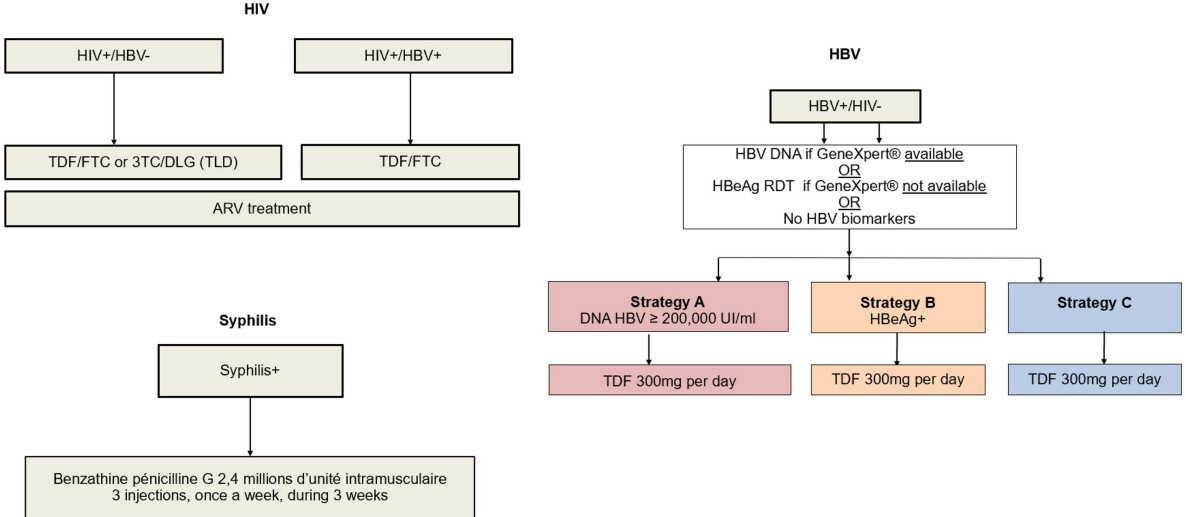

**Fig 2. Algorithm for HIV, syphilis and HBV treatment in pregnant women.** TDF: tenofovir disoproxyl fumarate; FTC: emtricitabine; 3TC: lamivudine; DLG: dolutegravir; TLD: dolutegravir; HBeAg: Hepatitis B e antigen; VL: viral load; HBV: hepatitis B virus.



range of flexible measures, such as individual psychosocial support interviews, focus group discussions, home visits and appointment reminders.

## Study procedures and data collection

• Before and after cross-sectional mixed-methods study

In the absence of control study sites, the before and after cross-sectional study will feed specific objective 1 by allowing allow to assess the triple screening coverage in pregnant women before and after the intervention in the study sites (primary endpoint). It will further allow to meet specific objectives 3, 4 and 5 of the study objectives, to identify associated facilitators and barriers to antenatal triple screening, and to adherence to care for women in need of treatment, assess the health capabilities and vulnerabilities of women of childbearing age, and evaluate the acceptability of the strategy among pregnant women, their partners, healthcare workers and key stakeholders involved in PMTCT.

The cross-sectional study will be conducted both before and after the implementation of the strategy (approximately 18 months following its initiation), using the same methodology.

The quantitative survey will be conducted among:

• i) a random sample of women who delivered in the past four months and who attended one of the study sites for postnatal or immunization services (n = 800 in Burkina Faso and 600 in The Gambia). The sample size was determined to demonstrate a significant increase (with a power of >95%) in access to screening for all three infections, i.e., an increase of 5% to 80% for screening for all three infections among pregnant women attending their first antenatal clinic. The sample size will also provide sufficient power to study in detail the socio-economic and behavioural factors associated with non-use of screening for the three infections among pregnant women.

• ii) all healthcare workers and community staff involved in prenatal care and/or maternal and child health and PMTCT activities in the study sites (n = 150). These estimates correspond to an exhaustive sample (all caregivers and community personnel involved in prenatal care and maternal and child care activities are eligible).

Quantitative data collection will be carried out by trained fieldworkers using standardised questionnaires administrated face-to-face in the participant's preferred language (local language, French or English) in a space that guarantees confidentiality. Data will be pseudonymised and collected onto electronic questionnaires using the REDCap software. The women 's questionnaire will cover socio-economic, behavioural and psychosocial aspects (Table 3). The questionnaire for healthcare workers and community staff" will focus on professional details, knowledge about the three infections, training needs, and preferences and perceptions of the integrated PMTCT strategy (Table 3).

The qualitative survey, led by researchers and research assistants in social anthropology will be conducted among women attending maternal and child health services, their spouses/partners, the healthcare and community workers, and representatives of health authorities. It will encompass in-depth interviews, focus group discussions, and observations in the study sites.

In-depth interviews will be conducted using a semi-structured guide in a confidential setting (Table 3). Focus groups will be organised at the participants' workplaces, or nearby. Observations will be conducted in the healthcare services using an observation grid to document structural constraints to screening and PMTCT activities (lack of resources, time, equipment, training), caregiver-patient relationships, and the quality of the information delivered during the pre- and post-test counselling sessions.

• intervention study with longitudinal follow-up study of pregnant women tested positive for any of the 3 infections

The intervention study with longitudinal follow-up relates to the global objective for the evaluation, and to specific objectives 1, 2 and 3: it will document treatment coverage among women screened positive for any of the three infections, i.e.,



**Table 3. Data collected in the cross-sectional study before and after the strategy implementation.**

| Data collected | Before | After |
|---|---|---|
| **1. Quantitative survey** | | |
| *Women who have recently given birth (<4 months)* | | |
| Socio-demographic and economic characteristics | X | X |
| Social and family environment, experiences of discrimination/violence | X | X |
| Mother and newborn health status and perceived health | X | X |
| Knowledge of the three infections and PMTCT and risk perception | X | X |
| Pregnancy follow-up and childbirth: pregnancy history, circumstances of the last pregnancy, follow-up and circumstances of childbirth | X | X |
| Screening and if positive, PMTCT measures received during pregnancy | X | X |
| High-risk sexual behaviours | X | |
| Psychological and behavioural characteristics (self-reported and revealed) | X | |
| Preferences regarding PMTCT | X | |
| **Healthcare workers and community staff** | | |
| Socio-demographic characteristics | X | X |
| Training and experience | X | X |
| Knowledge of the three infections and PMTCT and risk perception | X | X |
| Preferences regarding PMTCT | X | X |
| Challenges encountered in PMTCT and perceptions of the TRI-MOM strategy | X | X |
| **2. Qualitative survey** | | |
| **Women and partners (if the woman agrees)** | | |
| Experiences of ANC services and perception of service quality | X | X |
| Experience, perception, and acceptability of screening for the three infections | X | X |
| Beliefs, knowledge, and perception of health risks related to the 3 infections | X | X |
| Beliefs/perceptions about one's ability to preserve their health and that of their child/self-efficacy | X | X |
| *Women tested positive for at least one of the three infections* | | |
| Individual care pathways following positive screening, experience and management of the disease | X | X |
| Experiences of discrimination, rejection, violence | X | X |
| Experience and outcomes (achievements/failures) of capacity-building programs | | X |
| Experiences of care/treatment abandonment | X | X |
| **Healthcare workers and community staff** | | |
| Challenges encountered in offering and conducting screening for the 3 infections | X | X |
| Challenges encountered in case of positive screening (for counselling, referral, PMTCT) | X | X |
| Knowledge of the 3 infections and PMTCT | X | X |
| Perception and acceptability of the integrated screening approach | X | X |
| Experience (achievements/failures) of the integrated screening approach and capacity-building program | | X |
| **Representatives of national health authorities and programs** | | |
| Acceptability of the TRI-MOM strategy | X | X |
| Experience (achievements/failures) of the TRI-MOM strategy | | X |
| Skills, motivations to expand the TRI-MOM strategy to other sites | | X |

Abbreviations: PMTCT: prevention of mother-to-child transmission.

assessing: i) the antenatal cascade of care, including screening, treatment initiation, virological suppression or cure at delivery, ii) HepB-BD vaccine coverage in infants born to HBsAg-positive mothers, and iii) the acceptability of immediate treatment initiation in pregnant women.

Participation in the intervention study with longitudinal follow-up will be offered to all pregnant women screened positive for any of the three infections during their first antenatal care consultation at the study sites. These women will be enrolled at their first ANC visit and subsequently followed up at their 2nd, 3rd and 4th ANC visits until delivery (end of phase 1 follow-up).

Biological and clinical data will be collected on electronic tablets using standardized medical forms. Pre-inclusion and enrolment questionnaires will capture socio-demographic characteristics, testing results, pregnancy outcomes, clinical parameters of the consultation, biological tests results (viral load for HIV and HBV infections if available) and treatments. Follow-up visits will document the main clinical parameters of the pregnancy, pregnancy complications, and treatment adherence. The questionnaire at delivery will document pregnancy outcomes and the health status of the newborn (See Table 4).

Additionally, 7mL of peripheral blood using 1 EDTA tube and 1 card of dried blood spots (DBS) will be collected from all participating women at enrolment and delivery. The EDTA 7mL tube will be used for plasma storage (after centrifugation) at −20°C degrees and DBS will be stored at room temperature before being transported to the Centre Muraz in Burkina Faso and MRCG in The Gambia for storage at local biobanks.

Approximately 17,000 pregnant women are expected to be screened in the study sites during the implementation period (mid-2024 to mid-2026). This estimate is based on the number of first ANC visits performed monthly at the study sites and assumptions about screening uptake rates. Given the prevalence estimates, it is expected that approximately

**Table 4. Summary of data collected during the intervention study with longitudinal follow-up of pregnant women tested positive for any of the three infections.**

| Categories of data collection | ANC 1 | ANC 2 | ANC 3 | ANC 4 | Delivery |
|---|---|---|---|---|---|
| **1-Screening questionnaire** | | | | | |
| Data based on the antenatal card | X | | | | |
| Screening of the three infections offered/ accepted/ carried out | X | | | | |
| In case of screening refusal: reasons; consultation with peer mediator offered/accepted | X | | | | |
| In case of uptake to screening: test results for the three infections | X | | | | |
| **2-Enrolment questionnaire** | | | | | |
| Physical examination<br>Collection of blood samples in infected mothers (routine and biobank) | X | | | | X |
| Ongoing treatments | X | | | | |
| **3-Follow-up questionnaire** | | | | | |
| Physical examination | | X | X | X | |
| Adherence to HIV/ syphilis/ HBV treatment | | X | X | X | |
| **4-Delivery questionnaire** | | | | | |
| Physical examination<br>Collection of routine samples in HBV and HIV-infected mothers (routine and biobank) | | | | | X |
| Adherence to HIV/ syphilis/ HBV treatment | | | | | X |
| Delivery: date, place, complications<br>HBV and HIV viral load level | | | | | X |
| Child's health status at birth, treatments, and HepB-BD vaccine administration | | | | | X |

Abbreviations: ANC: antenatal care; HBsAg: Hepatitis B virus surface antigen; HBV: hepatitis B virus; HepB-BD: hepatitis B birth dose; MTCT: mother-to-child transmission.

2,365 women will be screened positive to at least one of the three infections across both countries, with 1,158 testing positive for HBV [17,30].

• Costing study

A micro-costing study will be conducted at each study site to estimate the costs of the strategy compared to the standard of care prior to its implementation. Data will be collected using standardised forms to document: i) quantities and unit costs of medical resources used for the screening and PMTCT of the three infections, including biologicals tests, treatments, and vaccines, ii) the number of clinical consultations during pregnancy and associated fees, iii) qualifications, salaries and time invested by the staff involved throughout the care process, from screening to delivery, and iv) medical resources required for managing any complications related to the three infections and their corresponding unit costs.

## Outcomes

The primary outcome of the study will be the proportion of women screened for the three infections in the study sites, as assessed in the "before" and "after" cross sectional studies.
Secondary outcomes include:
*Acceptability to the triple screening (cross-sectional study):*

• proportion of women who agreed to be screened for the three infections, as well as for each individual infection

• proportion of women who know their serological status for the three infections, and for each individual infection

• socio-economic, psychosocial and behavioural factors associated with screening uptake and awareness of serological status

**Health capabilities and vulnerabilities of women of childbearing age (cross-sectional study)**

• proportion of women living in poverty (assessed using the household expenditures and household assets) and/or experiencing food insecurity

• proportion of women with low knowledge levels regarding the three infections and PMTCT measures

• proportion of women with low levels of health literacy and agency

• proportion of women experiencing depression

• proportion of women experiencing domestic violence and stigma

• profiles of socioeconomic and psychosocial vulnerabilities

**Cascade of care in women screened positive and their children (intervention study with longitudinal follow-up)**

• proportion of HBsAg-positive women assessed for HBV MTCT risk

• proportion of women at risk of MTCT who initiated and completed recommended treatment

• proportion of women at risk of MTCT who achieved virological suppression (for HIV and HBV infections) or cure (for syphilis) at delivery

• proportion of newborns who received the timely HepB-BD vaccine, i.e., within the 24 hours of birth

• factors associated with the uptake of PMTCT measures

**Acceptability to immediate treatment (intervention study with longitudinal follow-up)**

- proportion of women screened positive who initiated immediate treatment (i.e., same day initiation)

- proportion of women who adhered to PMTCT measures

- proportion of women retained in care (at least four ANC consultations during pregnancy)

- perceptions regarding the three infections and the triple elimination strategy

- barriers and facilitators of triple screening, treatment initiation, adherence and retention into care

**Costs and cost-effectiveness (costing study)**

- Average annual costs per participant, average number of infections diagnosed, average number of infections correctly treated, and number of new infections avoided in newborns with the triple elimination strategy compared to the current situation

- Incremental cost-effectiveness ratio (ICER) of the triple elimination strategy versus the standard of care prior the intervention

## Statistical analysis

- "Before" and "after" cross-sectional study

Overall participation in each quantitative survey will be described using participation rates, and a comparison of the main sociodemographic characteristics between participants and non-participants will be conducted to identify potential selection bias. The study population will be described in terms of sociodemographic and economic characteristics. For variables with missing values exceeding 10% multiple imputation method will be employed for covariates, and Heckman-type models will be utilised for outcomes. Participant's characteristics will be compared between the "before" and "after" surveys, using chi-square tests for categorical variables and Mann-Whitney-Wilcoxon tests for continuous variables. In case of significant differences, inverse-probability weighting will be employed to reduce covariate imbalances between participants in the "after" survey (who benefited from the strategy) and participants in the "before" survey (who did not benefit from the strategy) [31]. The primary outcome will be computed for both the "before" and "after" surveys, and a probit regression model will be conducted to identify factors associated with triple screening and more specifically the effect of the strategy. A similar approach will be used to assess factors associated with the secondary outcomes and the effect of the strategy.

For the qualitative survey, the Word files containing transcriptions will be analysed manually (thematic analysis) and/or using the IramuteQ software. Thematic analysis will be conducted using a combined deductive and inductive approach.

- intervention study with longitudinal follow-up

The main sociodemographic and clinical characteristics of the participants will be described at baseline and at each visit to identify potential selection bias due to attrition or intermittently missed visits. Descriptive statistics of care cascade outcomes will be provided at the relevant time points, with temporal trends described when multiple time points are available (e.g., for adherence and retention). Multivariable random-effects panel data models will be used to identify the factors associated with the different outcomes. Random effects generalised linear and probit models will be used for continuous and binary outcomes, respectively.



- Costs and cost-effectiveness analysis

Costs will be estimated using a modified societal approach that incorporates all costs, irrespective of the entity bearing them. Outcomes will be estimated over the follow-up period for both the new strategy and the reference strategy to calculate the difference in costs between the two (incremental cost) as well as the difference in effectiveness (new HBV infections prevented in new-borns, estimated using the mother's biological characteristics and risk of MTCT at delivery). The analysis will also disaggregate the different health benefits acquired and the associated costs based on the socio-economic status of the mother (extended cost-effectiveness analysis). Incremental cost-effectiveness ratios will be compared against the cost-effectiveness thresholds of the study countries. Sensitivity analyses will be conducted.

All analyses will be performed using the Stata/SE 17.0 for Windows and RStudio 2023.12.1 for Windows.

## Status and timeline of the study

Participants recruitment will be completed by February 2026. Data collection will be completed by August 2026 and results are expected by June 2027.

## Ethics and dissemination

- Ethical consideration

The study protocol has been approved by the LSHTM/MRCUG Scientific Coordinating Committee (reference 28524), the Gambia Government/MRC Joint Ethics Committee (reference 28524), the LSHTM ethics committee (reference 28524), the Burkinabese National Ethical Committee for Research in Health (deliberation n° 2023-02-17) and the French Commission on Information Technology and Liberties (reference DR923079).

- Information, consent and data confidentiality

All participants enrolled in the project will receive full information on the objectives, the implications and the foreseeable constraints and risks of the research and will sign informed consent form. Quantitative and qualitative data will be pseudonymised using a unique study ID per participant. Pseudonymised questionnaires will be transmitted by secured connection to the data management centre (provided by the Medical Research Council Unit, The Gambia) and the final database transferred for analysis to the research partners.

- Expected benefits and risks

The primary risk for participants revolves around the potential psychosocial impact of testing positive for infections. This risk will be mitigated through various measures provided by peer mediators as part of the women's empowerment program, including counseling and psychosocial support.

The expected benefits of participation include free testing and access to PMTCT measures for all three infections, as well as support in accessing long-term care for HIV and HBV infection. Following delivery, all women screened positive for any of the 3 infections and their newborns will continue followed-up within the phase 2 TRI-MOM project until the infants' 9 months of age. For the subset of women who initiated TDF during pregnancy, TDF will be interrupted 2-month postpartum and women will undergo reassessment for hepatic flare. In cases of sever flare, defined by ALT > 10 ULN, TDF will be reintroduced. Furthermore, all infants will be screened for HIV and HBV at 9 months of age. Then, women living with HIV and HBV and their exposed-newborns will receive continued followed-up within the national HIV program and/ or specialised services for hepatitis, accordingly. Syphilis-exposed children will also be referred to specialised services for appropriate care.



## Expected impact and dissemination plan

We will provide evidence about the feasibility of implementing a simplified and integrated strategy to support the triple elimination agenda in Africa. It is likely that our data will be replicable in other African countries at least in the subregion. Furthermore, additional evidence on the effectiveness of the stratety on the reduction of MTCT risk will be provided by the project's second phase.

Upon completion of the project, research findings will be shared and discussed during workshops held in both countries with the main stakeholders involved in the PMTCT of the three infections, including community, members healthcare providers, and national health authorities and programmes. In order to guide countries in transitioning towards integrating PMTCT of the three infections into ANC services, the findings will be disseminated across various platforms and to diverse audiences using different media. These include posters and leaflets for community members, workshops and briefing notes for decision-makers, presentations at international conferences, and publication of scientific articles in international journals.

However, there are also important differences across regions, such as variations in HIV prevalence, HBV vaccination coverage, and health service delivery models, which may limit direct generalisability of our results. Therefore, while our study will provide valuable insights for the broader "triple elimination" agenda in Africa, further implementation research in diverse epidemiological and health system contexts will be required to identify specific barriers and facilitators in other regions, particularly in East and Southern Africa.

## Conclusion

HBV PMTCT has been long neglected in Africa. The TRI-MOM project will address multiple research gaps on HBV PMTCT and on triple elimination of HIV, Syphilis and HBV MTCT in Africa. This project has a huge potential to support a next generation of African infants free of HIV, Syphilis and HBV infections.

---

### Strengths and limitations of the study

- Despite the existence of rapid screening tests and prophylactic treatments, mother-to-child transmission of HIV, Syphilis and HBV persists in both The Gambia and Burkina Faso because coverage rates of screening for the three infections during pregnancy are low.

- Implementing the triple elimination strategy in close collaboration with the national health authorities and programs will ensure the sustainability of PMTCT of HIV, syphilis, and HBV in pregnant women.

- Promoting an interdisciplinary approach to evaluate the triple strategy, using both quantitative and qualitative methods, will provide a comprehensive understanding of the acceptability and feasibility of the strategy in both African countries.

- Including an economic evaluation will provide decision-makers with key information on the costs and cost-effectiveness of the triple elimination strategy compared to standard of care. Such information is essential for the nation-wide implementation of the strategy.

- The first phase of the TRI-MOM project will only focus on the prenatal period until delivery and will therefore not provide information on the reduction of MTCT of the three infections. This outcome will be evaluated during a second phase of the project (2025–2028).

---



## Supporting information

**S1 Fig. Algorithm for HIV, syphilis, and HBV screening in pregnant women.** This figure presents the diagnostic algorithm used for prenatal screening of HIV, syphilis, and hepatitis B virus (HBV), including initial testing and subsequent confirmatory steps. Note: RDT = rapid diagnostic test. In case of any ambiguous RDT result, the test should be repeated. (PPTX)

## Author contributions

**Writing – original draft:** Erwan Vo Quang, Alice N Guingané, Lauren Périères, Gibril Ndow, Asta Jobe, Yusuke Shimakawa, Maud Lemoine, Sylvie Boyer.

**Writing – review & editing:** Erwan Vo Quang, Alice N Guingané, Lauren Périères, Gibril Ndow, Victor Some, Sheriff Badjie, Asta Jobe, Clarisse Gouem, Yusuke Shimakawa, Dramane Kania, Maud Lemoine, Sylvie Boyer.

## Acknowledgments

We thank the French National Research Institute for Sustainable Development (Institut de Recherche pour le Développement), the sponsor of the TRI-MOM project. We also thank Alice Cardon for designing the healthcare worker questionnaire and Sandie-Marie Szawlowski for her assistance with programming the healthcare worker questionnaire and verifying the programming of the cross-sectional survey.

TRI-MOM study group (alphabetical order):

• MRC Unit The Gambia: Bangura Rohey, Bah Souleyman, Dondeh Bai-Lamin, Bojang Lamin, Bojang Masey, D'Alessandro Umberto, Drammeh Sainabou, Dibba Bakary, Foon Issatou, Gando Bah Momodou, Lemoine Maud, Ndow Gibril, Jatta Abdoulie, Jallow E Abdoulie, Jammeh Kitabu, Samateh Yousoupha, Vo-Quang Erwan

• Young Gambian Mums Fund: Jobe Asta, Sallah Amie

• National AIDS Control Program: Badjie Sheriff, Bah Pa Ousman, Cham Fatou

• Director of health services: Bittaye Mustapha

• Centre Muraz/INSP: Gouem Clarisse, Guingane Alice Nanelin, Kania Dramane, Ouedraogo T.w Inoussa, Some Julie, Taofiki Ajani Ousmane, Touro Siaka

• REVS PLUS: Kambire Eve Arlette, Maré Daouda, Some Victor

• Institut Pasteur Paris: Shimakawa Yusuke, Vincent Jeanne Perpétue

• UMR 1252 SESSTIM: Boyer Sylvie, Bureau-Stoltmann Morgane, Carrieri Patrizia, Cissé Bakary, Coste Marion, Di Beo Vincent, Drabo Seydou, Fiorentino Marion, Ismail Mariam, Lacombe Antoine, Maradan Gwenaelle, Marcellin Fabienne, Périères Lauren, Petitfour Laurène, Protiere Christel, Protopopescu Camelia

• UMR 151 LPED: Ouattara Fatoumata

• Institute for Global Health, UCL: Lépine Aurélia, Szawlowski Sandie-Marie

• IRD: Cournil Amandine, Paszkiewicz Brune

• Expertise France: Audemard Candice, Da Silva Yolande, Favre Matthieu, Rivière Cécile

• TRI-MOM Scientific Committee: Cham Muhammadu Kabir (Gambia Medical and Dental Council), Jaiteh Fatou (MRCU The Gambia), Koueta Fla (Centre Hospitalier Universitaire Pédiatrique Charles-de-Gaulle, Ouagadougou), Larmarange Joseph (CEPED), Leroy Valériane (Inserm U1027), Noseda Veronica (Expertise France), Mayaud Philippe (LSHTM), Lesi Olufunmilayo (WHO), Orne-Gliemann Joanna (ISPED), Préau Marie (Université Lyon 2), Ségéral Olivier (ANRS Cambodge), Sombie Roger (CHU Yalgado Ouédraogo), Van de Perre Philippe (UMR 1058), Tuaillon Edouard (Université Montpellier)

**Disclaimer:** The funder of the study has had no role in study design, and will have no role in data collection, data analysis, data interpretation, writing of the report or decision to submit. The content of this article is the sole responsibility of the study team and does not necessarily reflect the views and opinions of L'Initiative and Expertise France.



**Patient and public involvement:** Women affected by the three infections will not be directly involved in the development of the research questions, the design, recruitment, or implementation of the project. However, two national associations in Burkina Faso and in The Gambia are partners of the project and in charge of developing and implementing the women's empowerment programme using a participatory community-based approach. Furthermore, public health leaders will be closely associated to the implementation of the project through their participation to key activities including training and capitalisation as well as through their participation to the steering committee of the project.

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
