## [Decision Letter · Decision Letter 0]

15 Jul 2025

Dear Dr. Vo Quang,

Thank you for submitting your manuscript to PLOS ONE. After careful consideration, we feel that it has merit but does not fully meet PLOS ONE’s publication criteria as it currently stands. Therefore, we invite you to submit a revised version of the manuscript that addresses the points raised during the review process.

We look forward to receiving your revised manuscript.

Kind regards,

Angelica Espinosa Miranda, M.D., Ph.D.

Academic Editor

PLOS ONE

Journal Requirements:

"I have read the journal's policy and the authors of this manuscript have the following competing interests: GN, and YS report research funding and consultancy fees from Gilead Sciences. ML reports consultancy fees from Abbott USA and Gilead Sciences. All other authors declare no competing interests."

4. We note that the original protocol that you have uploaded as a Supporting Information file contains an institutional logo. As this logo is likely copyrighted, we ask that you please remove it from this file and upload an updated version upon resubmission.

5. Please provide a complete Data Availability Statement in the submission form, ensuring you include all necessary access information or a reason for why you are unable to make your data freely accessible. If your research concerns only data provided within your submission, please write "All data are in the manuscript and/or supporting information files" as your Data Availability Statement.

6. Please ensure that you refer to Figure 3 in your text as, if accepted, production will need this reference to link the reader to the figure.

Additional Editor Comments:

The article is relevant and aligns with the WHO’s strategy for the triple elimination of mother-to-child transmission (MTCT) of HIV, syphilis, and hepatitis B. It presents an interesting approach to public policy development in Sub-Saharan African countries. However, the authors should clarify some points in the manuscript. More detailed information is needed on the MTCT situation in the two countries studied—specifically, the challenges and opportunities, as well as the similarities and differences between them. It is also important to explain the rationale for presenting aggregated indicators for the three infections, rather than analyzing the outcomes for each infection separately. Providing more detail about the selected study sites would also help readers better understand the context and the relevance of the findings.

Reviewers' comments:

Reviewer's Responses to Questions

**Comments to the Author**

1. Does the manuscript provide a valid rationale for the proposed study, with clearly identified and justified research questions?

Reviewer #1: Partly

Reviewer #2: Yes

2. Is the protocol technically sound and planned in a manner that will lead to a meaningful outcome and allow testing the stated hypotheses?

Reviewer #1: Partly

Reviewer #2: Yes

3. Is the methodology feasible and described in sufficient detail to allow the work to be replicable?

Reviewer #1: Yes

Reviewer #2: Yes

4. Have the authors described where all data underlying the findings will be made available when the study is complete?

Reviewer #1: No

Reviewer #2: Yes

5. Is the manuscript presented in an intelligible fashion and written in standard English?

Reviewer #1: Yes

Reviewer #2: Yes

You may also provide optional suggestions and comments to authors that they might find helpful in planning their study.

Reviewer #1: The authors present a summary of their protocol for a study to be conducted in Burkina Faso and the Gambia that will implement a model for integrated clinical services to improve prevention of vertical transmission of HIV, congenital syphilis, and hep B. The aim is for the study findings to inform broader efforts in the countries, region, and Africa to achieve triple elimination of these diseases.

The paper is somewhat lacking in its understanding of how their study would fit into broader initiatives to achieve triple elimination. There is some overreach on how their study can inform scale-up, even locally. For example, while reported to include a mix of rural and urban sites, there are no rural sites in Burkina Faso. While this may be appropriate (given the discussion with the Ministry to select the sites), it would be useful to have more local context integrated into the text to explain why the site selection worked out this way. On the other hand, the actual individual enrollment numbers for the quantitative survey and cohort study are substantial. Additionally, while the inclusion of a variety of study designs is interesting, the depth of description of the mixed methods evaluation is lacking. Overall, the balance could be shifted more towards the 2 countries and less towards the “big picture” that is contingent on the study findings and experiences with actual implementation

Title

1. The reference to this being a protocol paper should appear in the title.

Abstract

2. Clarify what the cost and CEA will examine.

3. The text is unclear in how it specifies the sample size. It will be conducted in 5 sites in each country – not 10 sites in each of both countries.

Strengths and limitations

4. Specify the country contexts.

5. The timeframe for the second phase is provided as starting in 2025 when phase 1 is being presented in this paper under review in 2025. This is confusing. And will there be another paper for phase 2? Please clarify.

6. The study site sample of 5 per country, split between rural and urban settings may not necessarily be considered “several” if looked at from within individual categories.

Introduction

7. It is unusual to have this section start with sub-headers.

8. Please use person-centered language throughout the entire manuscript and supplemental files. For example, “adolescents infected with HIV” should be adolescents acquiring HIV.

9. The epi data are out of date (e.g., 2018). Please update all metrics where such data are available.

10. The text emphasizes the general global situation more than the local context in these 2 countries. Please refocus and provide a succinct overview of the triple elimination targets before describing the situation with burden of disease and prevention access and uptake in the 2 countries.

11. With regards to hep B elimination, the most critical aspects are HBsAg screening of pregnant women and a country’s history of vaccination – whether the 3-dose series or birth dose implementation. Local vaccination practices are not mentioned until the end of the section and then in insufficient detail. For example, birth dose is not described in terms of the targeted and universal approaches referenced in WHO triple elimination validation criteria.

12. Overall, the section refers more generally to preventing these 3 infections and is lacking in its awareness of the WHO’s triple elimination strategy.

Methods

13. It would be clearer if the hypothesis appeared earlier in the manuscript and was integrated into the objectives.

14. It is somewhat confusing to have objectives in one section and then have them repeated or supplemented elsewhere in the methods.

15. Was there a sample size consideration for the mixed methods study? This is missing.

16. How will the mixed methods data be analysed? This is missing.

17. Suggest focusing the text on what the authors intend to inform by way of health practice and policy in the countries where the study will be conducted. While their future study findings can inform efforts in the region and the sub-continent, this is somewhat beyond the scope of the proposed study. It remains to be seen if and how the study data could/would be directly generalizeable to the “triple elimination agenda in Africa” as a whole.

18. Overall readability is limited. Although a protocol paper, it would be of greater interest if it added more information (through data, narrative explanation) for a reader who wanted to learn more about challenges and opportunities to achieve triple elimination targets in these 2 countries.

Reviewer #2: The article is highly relevant and aligns with Sustainable Development Goal 3, particularly with regard to the elimination of diseases such as vertically transmitted infections (HIV, syphilis, and hepatitis B).

It also proposes the integration of hepatitis B vertical transmission prevention strategies with existing—though not yet well implemented—initiatives aimed at preventing vertical transmission of HIV and syphilis. This is especially pertinent given that public policies addressing hepatitis B in Sub-Saharan African countries remain underdeveloped.

The implementation of the project described in the protocol has the potential to strengthen local efforts to eliminate vertical transmission and to enhance the maternal and child healthcare continuum.

The following points should be better explained:

- When will the Antenatal Care Panel test for HIV, syphilis, and hepatitis B surface antigen (HBsAg) (Abbott, USA) be used, and when will the SD BIOLINETM HIV/Syphilis Duo Test and DetermineTM HBsAg test be used?

- For what purpose, and for how long, will the biobank samples be stored?

It is also recommended to review the outcome indicators, especially when aggregating all infections into a single metric. Consider whether to maintain a composite indicator or to disaggregate outcomes by infection type. This would allow for a better understanding of the specific implementation aspects of the vertical transmission prevention protocol for each infection (e.g., the proportion of women at risk of MTCT who achieved virological suppression [for HIV and HBV infections] or cure [for syphilis] at delivery -Would it be more appropriate to report these infections jointly or separately?).

**Do you want your identity to be public for this peer review?** For information about this choice, including consent withdrawal, please see our Privacy Policy

Reviewer #1: No

Reviewer #2: No

---

## [Author Response · Author response to Decision Letter 1]

19 Nov 2025

We thank the editors for their review, and we respond point by point to the questions.

We checked the information provided in both Funding Information’ and ‘Financial Disclosure’ sections.

"I have read the journal's policy and the authors of this manuscript have the following competing interests: GN, and YS report research funding and consultancy fees from Gilead Sciences. ML reports consultancy fees from Abbott USA and Gilead Sciences. All other authors declare no competing interests."

We added this in the competing interests section.

We added the sentence into the text.

4. We note that the original protocol that you have uploaded as a Supporting Information file contains an institutional logo. As this logo is likely copyrighted, we ask that you please remove it from this file and upload an updated version upon resubmission.

We removed it from the file and uploaded an update version.

5. Please provide a complete Data Availability Statement in the submission form, ensuring you include all necessary access information or a reason for why you are unable to make your data freely accessible. If your research concerns only data provided within your submission, please write "All data are in the manuscript and/or supporting information files" as your Data Availability Statement.

As requested, we added a section about data availability statement.

6. Please ensure that you refer to Figure 3 in your text as, if accepted, production will need this reference to link the reader to the figure.

We thank the editor, and we added Figure 3 in the main text.

Additional Editor Comments:

The article is relevant and aligns with the WHO’s strategy for the triple elimination of mother-to-child transmission (MTCT) of HIV, syphilis, and hepatitis B. It presents an interesting approach to public policy development in Sub-Saharan African countries. However, the authors should clarify some points in the manuscript.

More detailed information is needed on the MTCT situation in the two countries studied—specifically, the challenges and opportunities, as well as the similarities and differences between them.

It is also important to explain the rationale for presenting aggregated indicators for the three infections, rather than analyzing the outcomes for each infection separately.

Providing more detail about the selected study sites would also help readers better understand the context and the relevance of the findings.

We agree with the editors, and we added MTCT epidemiology as well as challenges in both countries The Gambia and Burkina Faso.

Reviewers' comments:

Reviewer's Responses to Questions

Comments to the Author

1. Does the manuscript provide a valid rationale for the proposed study, with clearly identified and justified research questions?

Reviewer #1: Partly

Reviewer #2: Yes

2. Is the protocol technically sound and planned in a manner that will lead to a meaningful outcome and allow testing the stated hypotheses?

Reviewer #1: Partly

Reviewer #2: Yes

3. Is the methodology feasible and described in sufficient detail to allow the work to be replicable?

Reviewer #1: Yes

Reviewer #2: Yes

4. Have the authors described where all data underlying the findings will be made available when the study is complete?

Reviewer #1: No

Reviewer #2: Yes

5. Is the manuscript presented in an intelligible fashion and written in standard English?

Reviewer #1: Yes

Reviewer #2: Yes

6. Review Comments to the Author

You may also provide optional suggestions and comments to authors that they might find helpful in planning their study.

Reviewer #1: The authors present a summary of their protocol for a study to be conducted in Burkina Faso and the Gambia that will implement a model for integrated clinical services to improve prevention of vertical transmission of HIV, congenital syphilis, and hep B. The aim is for the study findings to inform broader efforts in the countries, region, and Africa to achieve triple elimination of these diseases.

The paper is somewhat lacking in its understanding of how their study would fit into broader initiatives to achieve triple elimination. There is some overreach on how their study can inform scale-up, even locally. For example, while reported to include a mix of rural and urban sites, there are no rural sites in Burkina Faso. While this may be appropriate (given the discussion with the Ministry to select the sites), it would be useful to have more local context integrated into the text to explain why the site selection worked out this way.

On the other hand, the actual individual enrollment numbers for the quantitative survey and cohort study are substantial. Additionally, while the inclusion of a variety of study designs is interesting, the depth of description of the mixed methods evaluation is lacking.

Overall, the balance could be shifted more towards the 2 countries and less towards the “big picture” that is contingent on the study findings and experiences with actual implementation

We thank the reviewer for this feedback, which made us enrich the presentation of what our project will add to the existing field of literature, and of the extent to which we expect to generalize our findings.

Title

1. The reference to this being a protocol paper should appear in the title.

We amended the title to explicit that this paper is a protocol.

Former version

Towards triple elimination of HIV, syphilis and HBV mother-to-child transmission: protocol of a simplified and integrated strategy in Burkina Faso and The Gambia: the TRI-MOM project, phase 1

New version

Towards triple elimination of HIV, syphilis and HBV mother-to-child transmission: protocol of a simplified and integrated strategy in Burkina Faso and The Gambia: protocol for the phase 1 of the TRI-MOM project

Abstract

2. Clarify what the cost and CEA will examine.

We expanded the corresponding paragraph of the abstract.

Former version

Methods and analysis: The strategy will be implemented in 10 rural and urban health facilities in both countries including four activities: i) training sessions for healthcare workers working in maternal and child health services, ii) screening of pregnant women of the three infections using rapid diagnostic tests at the first antenatal visit, iii) clinical assessment and treatment of women tested positive for any of the 3 infections, and iv) raising awareness on HIV, Syphilis and HBV PMTCT among pregnant women and empowering those screened positive. 17,000 pregnant women are expected to be screened. The strategy will be evaluated through an interdisciplinary, mixed-methods approach comprising three studies: i) a quantitative and qualitative cross-sectional study conducted both before and after the implementation of the strategy to assess its impact on triple screening coverage in pregnant women; ii) a cohort study of pregnant women positive for any of the three infections to assess the coverage of PMTCT measures; and iii) a cost and cost-effectiveness analysis.

New version

Methods and analysis: The strategy will be implemented in 10 rural and urban health facilities in both countries including four activities: i) training sessions for healthcare workers working in maternal and child health services, ii) screening of pregnant women of the three infections using rapid diagnostic tests at the first antenatal visit, iii) clinical assessment and treatment of women tested positive for any of the 3 infections, and iv) raising awareness on HIV, Syphilis and HBV PMTCT among pregnant women and empowering those screened positive. 17,000 pregnant women are expected to be screened. The strategy will be evaluated through an interdisciplinary, mixed-methods approach comprising three studies: i) a quantitative and qualitative cross-sectional study conducted both before and after the implementation of the strategy to assess its impact on triple screening coverage in pregnant women; ii) a cohort study of pregnant women positive for any of the three infections to assess the coverage of PMTCT measures; and iii) a cost and cost-effectiveness analysis of the project compared to the reference situation in each country, which will rely on a micro costing study to estimate the incremental cost of the strategy per mother/child couple compared with the reference situation in each country, and compare it to the number of avoided infections.

3. The text is unclear in how it specifies the sample size. It will be conducted in 5 sites in each country – not 10 sites in each of both countries.

We amended the test for more clarity.

Former version

The strategy will be implemented in 10 rural and urban health facilities in both countries

New version

The strategy will be implemented in 5 rural and urban health facilities in each country

Strengths and limitations

4. Specify the country contexts.

We amended the test for more clarity.

New bullet point

Despite the existence of rapid screening tests and prophylactic treatments, mother-to-child transmission of HIV, Syphilis and HBV persists in both The Gambia and Burkina Faso because coverage rates of screening for the three infections during pregnancy are low.

5. The timeframe for the second phase is provided as starting in 2025 when phase 1 is being presented in this paper under review in 2025. This is confusing. And will there be another paper for phase 2? Please clarify.

Phases 1 and 2 consist of distinct activities (Phase 1 focusing on the pregnancy and the delivery, Phase 2 starting from post-partum and lasting for 6 months for each baby) but there will be an overlap of them of the field, starting from the delivery of the first included woman. The delay for publication may have induced some confusion in the calendar.

The protocol of Phase 2 is at the core of another article, soon to be submitted.

6. The study site sample of 5 per country, split between rural and urban settings may not necessarily be considered “several” if looked at from within individual categories.

We replaced the corresponding bullet point with the first one on the context of Burkina Faso and The Gambia.

Introduction

7. It is unusual to have this section start with sub-headers.

We removed the first subheaders.

8. Please use person-centered language throughout the entire manuscript and supplemental files. For example, “adolescents infected with HIV” should be adolescents acquiring HIV.

We amended the manuscript following your guidance.

9. The epi data are out of date (e.g., 2018). Please update all metrics where such data are available.

We updated all possible data, when more recent was available.

10. The text emphasizes the general global situation more than the local context in these 2 countries. Please refocus and provide a succinct overview of the triple elimination targets before describing the situation with burden of disease and prevention access and uptake in the 2 countries.

We thank the reviewer for their coment, and improved the manuscript: we shortened general considerations on triple elimination, and expanded on the situation of Burkina Faso and The Gambia.

Former version

Despite the existence of rapid screening tests and prophylactic treatments, mother-to-child transmission of HIV, Syphilis and HBV persists in both The Gambia and Burkina Faso because coverage rates of screening for the three infections during pregnancy are low.

New version

Despite the existence of rapid screening tests and prophylactic treatments, mother-to-child transmission of HIV, Syphilis and HBV persists in both The Gambia and Burkina Faso because coverage rates of screening for the three infections during pregnancy are low.

11. With regards to hep B elimination, the most critical aspects are HBsAg screening of pregnant women and a country’s history of vaccination – whether the 3-dose series or birth dose implementation. Local vaccination practices are not mentioned until the end of the section and then in insufficient detail. For example, birth dose is not described in terms of the targeted and universal approaches referenced in WHO triple elimination validation criteria.

We expanded on these matters and specified the strategy adopted by both countries in terms of infant vaccine (universal in both countries)

12. Overall, the section refers more generally to preventing these 3 infections and is lacking in its awareness of the WHO’s triple elimination strategy.

We restructured the introduction and

---

## [Editor Report · Decision Letter 1]

14 Dec 2025

Towards triple elimination of HIV, syphilis and HBV mother-to-child transmission: protocol of a simplified and integrated strategy in Burkina Faso and The Gambia: protocol for the phase 1 of the TRI-MOM project

PONE-D-25-09454R1

Dear Dr. Quang,

We’re pleased to inform you that your manuscript has been judged scientifically suitable for publication and will be formally accepted for publication once it meets all outstanding technical requirements.

Kind regards,

Angelica Espinosa Miranda, M.D., Ph.D.

Academic Editor

PLOS One

Additional Editor Comments (optional):

The authors answered our comments and added all suggestions in the manuscript.
---

## [Editor Report · Acceptance letter]

PONE-D-25-09454R1

PLOS One

Dear Dr. Vo Quang,

I'm pleased to inform you that your manuscript has been deemed suitable for publication in PLOS One. Congratulations! Your manuscript is now being handed over to our production team.

Kind regards,

on behalf of

Dr. Angelica Espinosa Miranda

Academic Editor

PLOS One